# A Routing Optimization Method for LEO Satellite Networks with Stochastic Link Failure

Guohong Zhao, Zeyu Kang , Yixin Huang  and Shufan Wu *

The School of Aeronautics and Astronautics, Shanghai Jiao Tong University, Shanghai 200240, China;
ghzhao@sjtu.edu.cn (G.Z.); kangzeyu@sjtu.edu.cn (Z.K.); huangyxethan@sjtu.edu.cn (Y.H.)
* Correspondence: shufan.wu@sjtu.edu.cn; Tel.: +86-34208597

**Abstract:** In this paper, for an Low-Earth Orbit (LEO) satellite network with inter-satellite links, a routing optimization method is developed in the case of stochastic link failure. First, a discrete-time strategy is used for the satellite network to acquire several static topological graphs during a cycle. Based on the static topological graphs regarding stochastic link failure, a constraint model is established that constructs the task revenue, switching times and routing cost as indicators. Then, an improved Genetic Algorithm based on A* is proposed to optimize the topology under the constraint model. In particular, to reduce the cost of computation, a new generation strategy for the initial solution is presented which combines the roulette wheel operator and the A* algorithm. Finally, the effectiveness of the proposed method is illustrated by a group of numerical simulations for the network with stochastic link failure.

**Keywords:** routing optimization; stochastic link failure; LEO satellite networks



## 1. Introduction

With the development of communication technology, satellite internet has become an important part of the next generation of global communication systems [1,2], which provides internet service of low latency and high bandwidth (broadband) [3]. The United States, the European Union, Russia and China have developed their own satellite internet constellation (SIC) programs, such as Starlink and Telesat, which are being established [4,5].

The orbit of a satellite can be divided into High-Earth Orbit (HEO), Medium-Earth Orbit (MEO) and Low-Earth Orbit (LEO) in terms of altitude. Because of the benefits of low latency and the development of inter-satellite link (ISL) technology, the LEO satellite network has attracted much attention [6,7]. However, the topology of the network is much more complicated than that of HEO or MEO satellite networks due to the fact that the satellites in LEO move very fast [8]. Therefore, with a view to the characteristics of high-speed movement, the selection of an appropriate communication routing becomes the key of LEO satellite network data transmission.

At present, satellite routing algorithms can be divided into the Centralized Routing Algorithm (CRA) and Distributed Routing Algorithm (DRA) [9,10]. In CRA, the control center plans routing and sends results to the others in the network based on the collected status information of each satellite [11]. Furthermore, the topology within each time period is assumed static by dividing the time into several intervals, and then the shortest-route algorithms, such as Dijkstra's algorithm, are used for route planning in each time interval [12]. DRA is a connectionless routing algorithm that does not consider global nodes, reducing the requirement of memory space effectively. Nevertheless, the DRA requires high online processing capacity at each node and may not yield optimal solutions [13].

In addition, according to different goals, such as network connectivity, terminal utilization, average end-to-end distance, etc., different optimization methods will produce different network topologies [14,15]. In [16–18], for a satellite network using inter-satellite

laser link (ISLL), a random link allocation scheme was proposed, which randomly generates a connected network topology and has the goal of maximizing the number of links, showing that the logical connections to be formed depend on the line-of-sight visibility of satellites.

In terms of ISLL, due to the platform vibration, internal system noise and pointing error, there is a possibility of failure in satellite links [19]. In [20,21]; with the assumption of constant probability of failure and transmitted power, the modified Rayleigh method was used to calculate the pointing error and the optimal root-mean-square width of the Gaussian beam. In [22], based on the space optical communication link equation, the signal-to-noise ratio equations were given for the inter-satellite coherent optical receiving system with different aberrations.

In this paper, for an LEO satellite network, we establish the model with stochastic link failure, based on the task revenue, switching times and routing costs. To tackle this challenging problem, an improved Genetic Algorithm (GA) is proposed which changes the means of initial solution generation. The main contributions of this work are stated as follows:

(1)　Different from previous studies [23,24], we consider the case of ISLL failure by treating communication tasks as stochastic events and calculate the probability of failure in the network.

(2)　To optimize the topology of the network, the improved Genetic Algorithm based on A* (GA-AS) is proposed, including a new initial population generation strategy that produces initial solutions to provide an optimization direction, resulting in effectively reducing the computation cost.

The remainder of this paper is organized as follows. The LEO satellite network model, including optimization constraints and objective functions, is established with stochastic link failure in Section 2. In Section 3, the GA-AS is proposed and the improved A* strategy is explained in detail. Simulation results under different task scales and population size are presented in Section 4. Finally, conclusions are provided in Scetion 5.

## 2. Mathematical Model

### 2.1. LEO Satellite Network Formation

According to the area and function, the system can be divided into three parts: space, ground and user segment. In this paper, we assume that the space segment contains only satellites. The ground segment includes functional entities such as gateway stations and network management centers, and the user segment consists of various types of terminal equipment and application facilities.

The simplified model of the LEO satellite network is shown in Figure 1, which includes five satellites. Usually, the user segment selects the upper satellite (SA I) and communicates with it through microwave, and then transmits data to the satellite (SA V) above the ground station via the relay satellite through ISLL, and finally the SA V communicates with the ground segment via microwave. Since the system adopts ISLL, the quality and range of transmission are greatly improved compared to microwave. On the one hand, link failure may occur during the communication due to the transmission errors and system noise. On the other hand, the complexity of the topology will increase significantly with the number of satellites due to the adoption of ISL.

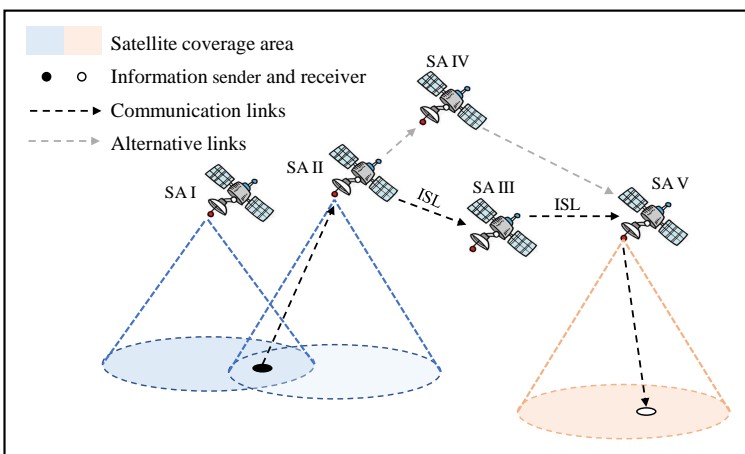

**Figure 1.** Simplified schematic diagram of Low-Earth Orbit (LEO) satellite network.

*2.2. Graph Theory*

Satellites' operation is cyclical, so the Discrete-Time Dynamic Virtual Topology Routing (DT-DVTR) method can be used to divide the time. In each time interval, as illustrated in Figure 2, the satellite network topology can be approximately regarded as static.

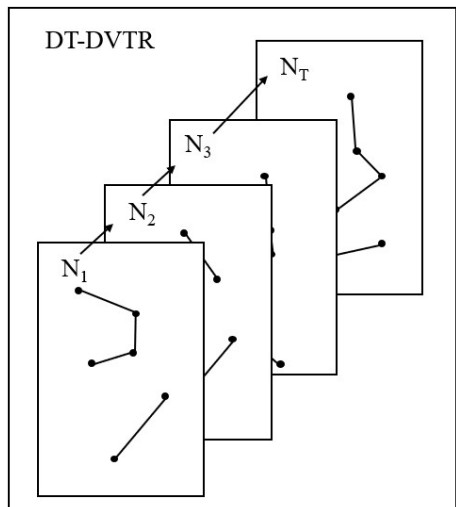

**Figure 2.** Key diagram of Discrete-Time Dynamic Virtual Topology Routing (DT-DVTR) algorithm.

Supposing a satellite network is divided into $N_T$ static topologies, which is composed of $n$ satellites in period $T$, the $c$th topology can be represented by a static directed graph $\mathcal{G}(c) = (\mathcal{V}, \mathcal{E}(c))$. Nonempty finite set $\mathcal{V}(c)= \{v_1, \cdots, v_n\}$ and $\mathcal{E}(c) \subseteq \mathcal{V} \times \mathcal{V}$ represent the node and edge set, respectively. Matrix $\mathcal{A}_c=[a_{ij}^c] \in \mathbb{R}^{n \times n}$, $a_{ij}^c \in \{0, 1\}$ represents the adjacency matrix if $(v_i, v_j) \in \mathcal{E}$, $a_{ij} = 1$ $((v_j, v_i) \in \mathcal{E}$, $a_{ji} = 1)$; otherwise, $a_{ij} = 0$ $(i, j = 1, \ldots, n)$. The adjacent set of node $v_i$ is defined as $\mathcal{N}_i = \{j|(v_i, v_j) \in \mathcal{E}\}$. The in-degree matrix of $\mathcal{G}(c)$ is denoted by $\mathcal{L} = \text{diag}\{l_1, \cdots, l_n\}$, where $l_i = \sum_{j \in \mathcal{N}_i} a_{ji}$, $i = 1, \cdots, n$ is the number of edges with $v_i$ as the end node, which is called the in-degree of $v_i$. Correspondingly, $\mathcal{O} = \text{diag}\{o_1, \cdots, o_n\}$ is defined as the out-degree matrix, where $o_i = \sum_{j \in \mathcal{N}_i} a_{ij}$, $i = 1, \cdots, n$ represents the number of edges with $v_i$ as the starting node, which is called the out-degree of $v_i$. The cost matrix is denoted by $\mathcal{D}^c = [d_{i,j}^c] \in \mathbb{R}^{n \times n}$, $d_{i,j}^c \in [0, +\infty)$, where $d_{i,j}^c$ represents the cost of $(v_i, v_j)$.

*2.3. LEO Satellite Network Model*

2.3.1. Communication Routing Model

Suppose $\mathcal{VP}^c = \{vp_1^c, \cdots, vp_{NS_c}^c\}$ is the set of communication tasks in $\mathcal{G}(c)$, and each task contains a start node $v_k^{c,start}$ and a end node $v_k^{c,end}$, which denoted as $vp_k^c = \{v_k^{c,start}, v_k^{c,end}\}$, $v_k^{c,start}, v_k^{c,end} \in \mathcal{V}$. We make the following assumptions:

- Assumption 1: For $vp_k^c$, there are $NM_k^c$ feasible paths $path_r^{c,k}$, $r = 1, \cdots, NM_k^c$ ($NM_k^c \geq 1$ means that there is at least one feasible path in $vp_k^c$). All feasible paths form the set $\mathcal{P}_k^c = \{path_1^{c,k}, \cdots, path_{NM_k^c}^{c,k}\}$.

- Assumption 2: An arbitrary feasible path $path_r^{c,k}$ consists of non-repeating edges. $\delta_{(i,j)}^{path_r^{c,k}} = 1$, $i, j = 1, \cdots, n$, $r = 1, \cdots, NM_k^c$ is used to represent $(v_i, v_j) \in path_r^{c,k}$, and vice versa.

The routing selection of $vp_k^c$ is equivalent to selecting a path with the least cost from the $NM_k^c$. Let $po_k^c$ be the optimal path in $\mathcal{P}_k^c$, and all optimal paths $po_k^c$, $k = 1, \cdots, NS_c$ form the optimal path set $\mathcal{PO}^c = \{po_1^c, \cdots, po_{NS_c}^c\}$. The optimal connection matrix of $\mathcal{G}(c)$ is represented by $\mathcal{H}^c = [h_{ij}^c] \in \mathbb{R}^{n \times n}$ $h_{ij}^c \in \{0, 1\}$ with the property that if $(v_i^c, v_j^c) \in \mathcal{PO}^c$, $h_{ij}^c = 1$; otherwise, $h_{ij}^c = 0 (i, j = 1, \cdots, n)$.

2.3.2. Link Switch Model

The satellite undergoes a series of works such as alignment before establishing the ISLL. It costs more and takes more time to establish new communications than to maintain the previous connection. We suppose that in any two adjacent $\mathcal{G}(c-1)$ and $\mathcal{G}(c)$, $c = 2, \cdots, N_T$, $\mathcal{PO}^{c-1}$ and $\mathcal{PO}^c$ are the optimal path sets, respectively, $\mathcal{H}^{c-1}$ and $\mathcal{H}^c$ are the corresponding optimal connection matrix, respectively. $\mathcal{NS}_c = [ns_{ij}^c] \in \mathbb{R}^{n \times n}$, $ns_{ij}^c \in \{0, 1\}$ is the switch matrix of $\mathcal{G}(c)$, and $ns_{ij}^c$ is defined as:

$$ns_{ij}^c = \begin{cases} 1, h_{ij}^c = 1, h_{ij}^{c-1} = 0 \\ 0, others \end{cases} c = 2, \ldots, N_T \tag{1}$$

In particular, we define $\mathcal{NS}_0 = \mathbf{0}$.

*2.4. Stochastic Link Failure Model*

2.4.1. Inter-Satellite Laser Link Signal (ISLL) Model

It is assumed that the ISLL system adopts Intensity Modulation Direct Detection and Gaussian beam with the direction from satellite $i$ to $j$. According to [22], the received luminous power $P_R$ of satellite $j$ can be expressed as:

$$P_R = P_T G_T G_R \eta_T \eta_R L_T \left(\frac{\lambda}{4\pi d_{ij}}\right)^2 \tag{2}$$

where $P_T$ is the emitted luminous power; $G_T$ is the emitted antenna gain; $G_R$ is the received antenna gain; $\eta_T$ and $\eta_R$ are the transmitting and receiving antenna efficiency, respectively; $L_T = \exp(-G_T \theta_{ij}^2)$ is the pointing error loss factor, where $\theta_{ij}$ is the pointing error deviation angle; $\lambda$ is the laser wavelength ; $d_{ij}$ is the transmission distance.

The detector of satellite $j$ converts the received luminous power $P_R$ into current signal $I_j$, which is expressed as:

$$I_j = RP_R \tag{3}$$

where $R$ is the responsivity. Combining Equations (2) and (3), we obtain:

$$I_j = RP_T G_T G_R \eta_T \eta_R \left(\frac{\lambda}{4\pi d_{ij}}\right)^2 \exp(-G_T \theta_{ij}{}^2) \tag{4}$$

### 2.4.2. Pointing Error

The beam radial pointing error angle of satellite $i$ is denoted by $\theta_{ij} = (\theta_x, \theta_y)^T$, where $\theta_x \sim N(\mu_x, \sigma_x^2)$ and $\theta_y \sim N(\mu_y, \sigma_y^2)$ represent the pointing error angle of azimuth and pitch, respectively. Thus, $\theta_{ij} = \sqrt{\theta_x^2 + \theta_y^2}$ obeys the Beckman distribution [25] and approximates it to the modified Rayleigh distribution with parameter $\sigma_{\text{mod}}$ [20]. The approximate expressions of the probability density function (PDF) and cumulative distribution function (CDF) of $\theta_{ij}$ are obtained as:

$$f_\theta(\theta_{ij}) \approx \frac{\theta_{ij}}{\sigma_{\text{mod}}^2} \exp\left(-\frac{\theta_{ij}{}^2}{2\sigma_{\text{mod}}^2}\right), \theta_{ij} \geq 0 \tag{5}$$

$$F_\theta(\theta_{ij}) \approx 1 - \exp\left(-\frac{\theta_{ij}{}^2}{2\sigma_{\text{mod}}^2}\right) \tag{6}$$

where

$$\sigma_{\text{mod}} = \left(\frac{3\mu_x^2 \sigma_x^4 + 3\mu_y^2 \sigma_y^4 + \sigma_x^6 + \sigma_y^6}{2}\right)^{1/6} \tag{7}$$

### 2.4.3. Stochastic Link Failure

Generally, when the channel capacity $C$ is insufficient to meet the requirement of transmission rate $R_0$, we refer to this situation as communication failure. The stochastic failure matrix is denoted by $\mathcal{B}^c = [b_{ij}^c] \in \mathbb{R}^{n \times n}$, where $b_{ij}^c \in [0, 1]$ represents the probability of communication failure from satellite $i$ to $j$, expressed as

$$b_{ij}^c = P[C(S) \leq R_0] \tag{8}$$

where $S$ denotes the signal-to-noise ratio (SNR). Since $C(\bullet)$ increases monotonically with $S$, the failure probability can be shown as

$$b_{ij}^c = P(S \leq a) \tag{9}$$

In the formula, $a$ is the preset threshold, and $C^{-1}(\bullet)$ is the inverse function of $C(\bullet)$.

At the receiving system, the signal has additive white Gaussian noise with mean of 0 and variance of $\sigma_N^2$ after the photoelectric detector. Therefore, the instantaneous signal-to-noise ratio $S$ of the electrical signal received can be represented as:

$$S = \frac{I^2}{2\sigma_N^2} \tag{10}$$

Combining Equations (4), (6) and (10), the CDF of $S$ can be indicated as

$$F_s(s) = P(S \leq s) = P\left[\theta \geq \sqrt{\frac{1}{2G_T} \ln \frac{B}{s}}\right] = 1 - F_\theta\left[\sqrt{\frac{1}{2G_T} \ln \frac{B}{s}}\right] = \frac{s}{B} \exp(\frac{1}{4G_T \sigma_{\text{mod}}^2}) \tag{11}$$

where

$$B = \frac{1}{2\sigma_N^2} \left(\frac{\lambda}{4\pi d_{ij}}\right)^4 (RP_T G_T G_R \eta_T \eta_R)^2 \tag{12}$$

For a given threshold $a$, the failure probability is calculated by

$$b_{ij}^c = P(S \leq a) = \frac{a}{B} \exp\left(\frac{1}{4G_T \sigma_{\text{mod}}^2}\right) \tag{13}$$

*2.5. Operational Constraints*

Based on the above discussion, we establish a multi-constraint satisfaction model of the LEO satellite network. Some assumptions are made as follows:

1. The on-board power consumption of satellites is only affected by the number of ISL;
2. Attenuation of microwave is ignored due to other reasons such as outside interference;
3. Once satellite communication starts, it cannot be interrupted;
4. During the entire period $N_T$, the quantity of satellites remains the same.

According to the characteristics and operational regulations of the satellite formation, some constraints can be listed as follows:

- (1) Communication constraints between satellite and gateway station.

If the $i$th satellite is visible to the $p$th gateway station, the angle $\alpha_{pi}$ between the vector of the satellite's centroid pointing to the gateway station and connecting the satellite to the center of the Earth cannot exceed the satellite's maximum offset capability $\alpha_{s\max}$, expressed as:

$$\alpha_{pi} \leq \alpha_{s\max} \tag{14}$$

- (2) Communication constraints between satellite and user terminals.

If the $i$th satellite is visible to the $q$th user, the angle $\alpha_{qi}$ between the vector of the satellite's centroid pointing to the terminal and connecting the satellite to the center of the Earth cannot exceed the satellite's maximum offset capability $\alpha_{o\max}$. It is expressed as:

$$\alpha_{qi} \leq \alpha_{o\max} \tag{15}$$

- (3) Satellite's microwave communication capability constraints.

When the satellite communicates with ground gateways and terminals, data uploads and downloads occur. The data upload speed $p_{up}^i$ and download speed $p_{down}^i$ of satellite $i$ cannot exceed the maximum values $p_{up\max}$ and $p_{down\max}$, expressed as:

$$p_{up}^i \leq p_{up\max} \tag{16}$$

$$p_{down}^i \leq p_{down\max} \tag{17}$$

- (4) Satellite laser communication capability constraints.

The number of inter-satellite links established by satellites at the same time is limited; that is, the out-degree $o_i$ and in-degree $l_i$ of the satellite $i$ cannot exceed their maximum values $o_{\max}^i$ and $l_{\max}^i$. This is expressed as:

$$o_i \leq o_{\max}^i \tag{18}$$

$$l_i \leq l_{\max}^i \tag{19}$$

- (5) ISLL communication range restriction.

The communication distance $d_{ij}$ between satellite $i$ and $j$ cannot exceed the maximum $d_{\max}$, expressed as:

$$d_{ij} \leq d_{\max} \tag{20}$$

*2.6. Objective Function*

The function $f_1$ represents the revenue of tasks, expressed as:

$$f_1 = \sum_{k=1}^{NS_c} \psi_k^c \, priority_k^c \tag{21}$$

where $priority_k^c$ represents the priority of task $vp_k^c$, and $\psi_k^c \in [0,1]$ represents the probability of $vp_k^c$ being executed. The calculation formula of $\psi_k^c$ is as follows:

$$\psi_k^c = \prod_{i,j} \left(1 - \delta_{(i,j)}^{po_k^c} b_{ij}^c\right) \tag{22}$$

Function $f_2$ represents the number of all new links in the graph $\mathcal{G}(c)$, expressed as:

$$f_2 = \sum_{i=1}^{n} \sum_{j=1}^{n} ns_{ij}^c \tag{23}$$

where $ns_{ij}^c$ is an element in $\mathcal{NS}_c$.

Function $f_3$ represents the total cost of the graph $\mathcal{G}(c)$, expressed as:

$$f_3 = \sum_{k=1}^{NS_c} \sum_{i,j} \delta_{(i,j)}^{po_k^c} d_{i,j}^c \tag{24}$$

Taking into account the different magnitudes and importance of $f_1$, $f_2$ and $f_3$, the non-dimensional parameter $\mathbf{q} = [q_1, q_2, q_3]^\top$ and weight $\mathbf{w} = [w_1, w_2, w_3]^\top$ are introduced into the objective function $f$. The optimization objective is defined as:

$$\max f = (w_1 \cdot p_1 \cdot f_1 - w_2 \cdot p_2 \cdot f_2 - w_3 \cdot p_3 \cdot f_3) \tag{25}$$

where $w_1, w_2, w_3 \in [0,1]$ represents the weight coefficient, set according to the actual situation.

## 3. Optimization Method

The complexity of the network topology increases dramatically with the number of satellites. On the one hand, it is hard to solve the multi-constraint problem by various shortest-route algorithms; on the other hand, the initial solution of intelligent optimization algorithms is generally generated by a stochastic approach, so there is a large uncertainty in the optimization direction. Based on the above considerations, we propose the improved GA algorithm based on A* (GA-AS), which introduces the roulette wheel selection into the A* algorithm to generate a series of solutions, and then the solutions are used as the initial population of the Genetic Algorithm (GA).

*3.1. Improved Genetic Algorithm Based on A* (GA-AS)*

GA is a self-organizing and self-adaptive artificial intelligence technology that simulates the evolutionary process and mechanism of natural organisms to solve problems. The basic framework of GA-AS is similar to GA's but the initial solution generator has been significantly modified. Here, we give the framework of GA-AS, and the flow of it is shown in Algorithm 1.

- Initialization: The initial population needs to be generated before the iterative calculation is performed. Assuming that the population size is $N$, we adopt the improved A* strategy and run the algorithm $N$ times to generate the initial population in this work.
- Coding: The coding is the process of establishing the mapping relationship between phenotype and genotype. We use natural number coding to transform the problem into an optimal combination problem.

- Selection: In this process, the principle of competition in nature is simulated by calculating the fitness value, which is the criterion for assessing individuals' performance, and then the outstanding individual is selected.
- Heredity: This part includes crossover and mutation. Inspired by biology, the offspring is produced by gene crossover and mutation of parents. Usually, the offspring shares almost features of the parents, and it is possible for the offspring to have properties that the parent does not have due to the mutation operator.

In GA-AS, when the genotypes change, the population needs to be filtered to screen out the individuals who violate the constraints. In this work, for addressing the situation, we use the penalty strategy by reducing the objective function value $f$ to obtain fitness value $\mathcal{F}$, as shown in Equation (26), and therefore $\mathcal{F}$ may be negative.

$$\mathcal{F} = f - n \cdot M \tag{26}$$

where $n$ is the number of constraint violations, and $M$ is a constant.

---

**Algorithm 1** Improved Genetic Algorithm Based on A* (GA-AS)

---

    **Input:** communication tasks
    **Output:** routing optimization results
1  initialization: generate initial population by the improved A* strategy;
2  **while** *iteration number less than the set value **or** average population fitness value greater than the threshold value* **do**
3     |   calculate the objective function value $f$;
4     |   count the number of constraint violations $n$ for per individual;
5     |   calculate fitness value $\mathcal{F}$ according to Equation (26);
6     |   heredity (crossover and mutation);
7     |   recalculate $\mathcal{F}$;
8     |   generate new population;
9     |   selection;
10 **end**

---

### 3.2. Improved A* (A-Star) Strategy

We introduce the roulette wheel selection into the A* algorithm to obtain the improved A* strategy. For different paths, the strategy making the high-cost paths has non-zero probability of selection, as shown in Figure 3. In $\mathcal{G}(c) = (\mathcal{V}, \mathcal{E}(c))$, $partpath_r^{c,k} \subseteq path_r^{c,k}$, $r = 1, \cdots, NM_k^c$ represents the partial of $path_r^{c,k}$, and $v_s^{c,k}$ denotes the end node that $partpath_r^{c,k}$ passes through. The $\mathcal{U}_s = \{v_{s_1}, \cdots, v_{s_m}\}, v_{s_i} \in \mathcal{V},\ i = 1, \cdots, m$ is called the reachable node set of $v_s^{c,k}$ when it satisfies the following conditions:

1. $\forall v_{s_i} \in \mathcal{U}_s$, satisfy $(v_{s_i}, v_s^{c,k}) \notin partpath_r^{c,k}$ and $s_i \in \mathcal{N}_s$.
2. $\forall v_i \notin \mathcal{U}_s$, satisfy $(v_i, v_s^{c,k}) \in partpath_r^{c,k}$ or $i \notin \mathcal{N}_s$.

The calculation formulas for the selection probability $P(v_{s_i})$ and cumulative probability $Q(v_{s_i})$ of any node $v_{s_i} \in \mathcal{U}_s$ are as follows:

$$P(v_{s_i}) = \frac{\frac{1}{h(v_{s_i})}}{\sum_{v_{s_i} \in \mathcal{U}_s} \frac{1}{h(v_{s_i})}}, i = 1, \cdots, m \tag{27}$$

$$Q(v_{s_i}) = \sum_{j=1}^{i} P(v_{s_i}) \quad i = 1, \cdots, m \tag{28}$$

$$h(v_{s_i}) = h_1(v_{s_i}) + h_2(v_{s_i}) \tag{29}$$

Among them, $h(v_{s_i})$ represents the estimated cost of $path_r^{c,k}$, $h_1(v_{s_i})$ represents the actual cost of $partpath_r^{c,k}$, and $h_2(v_{s_i})$ represents the estimated cost between $v_{s_i}$ and $v_q^{c,k}$. The calculation formula is:

$$h_1(v_{s_i}) = \sum_{i,j} \left( \delta_{(i,j)}^{partpath_r^{c,k}} d_{i,j}^c \right) \tag{30}$$

$$h_2(v_{s_i}) = d_{s_i,q}^c \tag{31}$$

In particular, define $Q(v_{s_0}) = 0$. When $v_{s_a} \in \mathcal{U}_s$ satisfies the following formula, choose $v_{s_a}$ as the next path node:

$$Q(v_{s_{a-1}}) < z_s < Q(v_{s_a}) \tag{32}$$

where the random variable $z_s \sim U(0,1)$.

Before implementation of the improved A* strategy, an Open List and a Closed List ought to be established, which are used to record the nodes that need to be and have been investigated, respectively. The flow of the improved A* strategy is shown in Algorithm 2.

---

**Algorithm 2** The improve A* strategy.

**Input:** start and end node
**Output:** routing connection
1  initialization: put the first node into the Open List;
2  **while** *the Open List is nonempty and $v_k^{c,end}$ is not in the Open List* **do**
3      generate $z_s$ and calculate the cumulative probability $Q(v_{s_i})$;
4      select $v_{s_a}$ from the Open List as the current node *a* according to Equation (32);
5      add *a* to the Close List and remove it from the Close List;
6      calculate the $\mathcal{U}_s$ of node *a* and generate a set of child nodes;
7      **for** *each child node b* **do**
8          **if** *b in the Close List* **then**
9              discard it;
10         **else**
11             **if** *b in the Open List* **then**
12                 **if** *the h value calculated by node a is smaller* **then**
13                     update *h* value and set parent node of *b* as *a*;
14                 **end**
15             **else**
16                 add *b* into the Open List and calculate its *h* value;
17                 set the parent node of *b* as *a*;
18             **end**
19         **end**
20     **end**
21 **end**

---

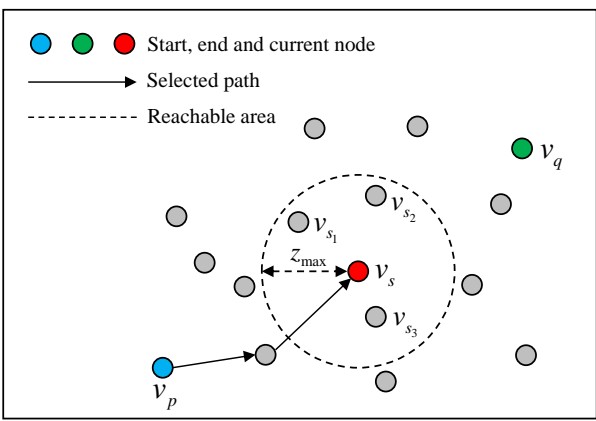

**Figure 3.** Improved A* strategy schematic diagram.

## 4. Simulation Experiment and Analysis

### 4.1. Stochastic Link Failure Experiment

This section analyzes the effects of transmission distance and transmitting power on the failure probability in ISLL systems. For each case, transmit gain $G_T$, receive gain $G_R$, transmit efficiency $\eta_T$, receive efficiency $\eta_R$, laser wavelength $\lambda$, responsivity $R$, noise variance $\sigma_N^2$ and threshold $a$ are set to $G_T = 106.3$ dB, $G_R = 118.9$ dB, $\eta_T = 0.5$, $\eta_R = 0.4$, $\lambda = 1064$ nm, $R = 0.6003$ A/W, $\sigma_N^2 = 4.3 \times 10^{-12}$ A$^2$, $a = 1 \times 10^{-6}$. Azimuth pointing error $\theta_x$ and pitch pointing error $\theta_y$ are set to $\theta_x \sim N(\mu_x, \sigma_x^2)$, $\theta_y \sim N(\mu_y, \sigma_y^2)$, respectively, where $\mu_x = \mu_y = 0$ and $\sigma_x = \sigma_y = 0.75$.

Table 1 lists the relationships between transmit power, transmission distance and failure probability under the above-mentioned condition. As can be seen from Table 1, in the condition of the same transmit power, the probability of failure is proportional to transmission distance, and it is very small and can be ignored, while the transmission distance is less than 1000 km. In addition, the transmission power is inversely proportional to the probability failure under the condition of the same transmission distance.

**Table 1.** Failure probability with different transmission distances and power.

| Transmit Power (mW) | Transmission Distance (km) | Probability of Failure |
|---|---|---|
| 10 | 500 | $2.85 \times 10^{-6}$ |
|  | 1000 | $4.56 \times 10^{-5}$ |
|  | 5000 | 0.03 |
|  | 10,000 | 0.45 |
| 30 | 500 | $7.13 \times 10^{-7}$ |
|  | 1000 | $1.14 \times 10^{-5}$ |
|  | 5000 | 0.01 |
|  | 10,000 | 0.11 |

### 4.2. Optimization Results

This section compares the experimental results of GA and GA-AS. The experiment randomly generates several tasks and environments, including satellites, user terminals and ground stations. The parameter settings are as follows, where mutation probability $\eta_m = 0.05$, genetic probability $\eta_g = 0.5$, crossover probability $\eta_c = 0.5$, weight $\mathbf{w} = [0.8, 0.1, 0.1]^\top$, transmission power $P_T = 10$ mW. Other parameter settings are consistent with Section 4.1.

Table 2 lists the optimization results of GA-AS with different task sizes under the condition of population size $NP = 700$. The final value refers to the final result obtained by the GA-AS. It can be seen from Table 2 that when the number of tasks $NS_c$ is set to 41, the average fitness value of the initial population is roughly 81% of the final value. As the task size grows, the distribution range and mean value decrease. When $NS_c = 98$, the fitness value of the initial population is distributed in 61%~88% of the final value, and the mean value is around 78% of it, indicating that the new initial-solution generation strategy can produce a good initial population.

**Table 2.** Initial population distribution with different scales.

|  | Mean Value | Max Value | Min Value | Final Value | Variance |
|---|---|---|---|---|---|
| $NS_c = 41$<br>$NA = 50$ | $1.93 \times 10^3$ | $2.12 \times 10^3$ | $1.72 \times 10^3$ | $2.38 \times 10^3$ | $3.58 \times 10^3$ |
| $NS_c = 98$<br>$NA = 100$ | $4.24 \times 10^3$ | $4.74 \times 10^3$ | $3.94 \times 10^3$ | $5.78 \times 10^3$ | $1.24 \times 10^4$ |
| $NS_c = 300$<br>$NA = 150$ | $1.30 \times 10^4$ | $1.48 \times 10^4$ | $1.04 \times 10^4$ | $1.68 \times 10^4$ | $2.28 \times 10^6$ |

Figure 4 and Table 3 list the optimization results of GA-AS and GA with different population sizes when $NS_c = 74$ and $NA = 40$. Since penalties were used in the experiment, the fitness values may be negative. It can be seen from Figure 4 and Table 3 that the optimization capability of GA-AS is significantly better than GA when $NP = 500$ and improves as the population size increases. When the $NP$ reaches 2000, it has little effect on the optimization of GA-AS to increase the $NP$ due to the quite good performance. However, GA has almost no optimization ability when $NP$ is less than 2000 and it reached 50,000, which can be considered undesirable. Moreover, the performance of GA-AS is always better than GA under the same population size.

Figure 5 shows the probability of task execution for GA and GA-AS for different population sizes with $NS_c = 74$, and the value is correlated with the stochastic link failure. For GA-AS, at the population size of 500, most task execution probabilities are greater than 0.8 and increase with $NP$. In contrast, at a population size of 500 in GA, most of the tasks could not be executed and the optimization performance was improved with increasing population size, but there was still a considerable gap between the GA and GA-AS results.

**Table 3.** Optimization results with different population sizes.

| Algorithm | Population Size | Final Value | Running Time (min) |
|---|---|---|---|
| GA[a] | 500 | $-3.2971 \times 10^3$ | 3 |
| | 2000 | 57.5916 | 12 |
| | 10,000 | $2.2956 \times 10^3$ | 16 |
| | 20,000 | $2.6959 \times 10^3$ | 37 |
| | 50,000 | $3.5768 \times 10^3$ | 73 |
| GA-AS | 500 | $4.0939 \times 10^3$ | 5 |
| | 2000 | $4.4124 \times 10^3$ | 10 |
| | 10,000 | $4.4947 \times 10^3$ | 27 |
| | 20,000 | $4.5559 \times 10^3$ | 49 |
| | 50,000 | $4.5599 \times 10^3$ | 95 |

[a] Genetic Algorithm.

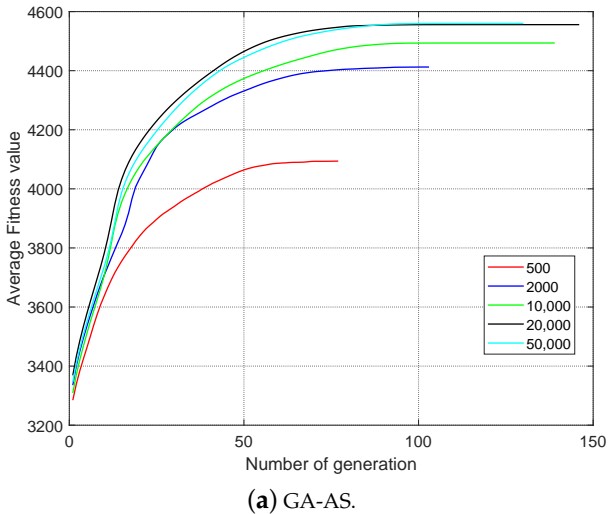

(**a**) GA-AS.

**Figure 4.** *Cont.*

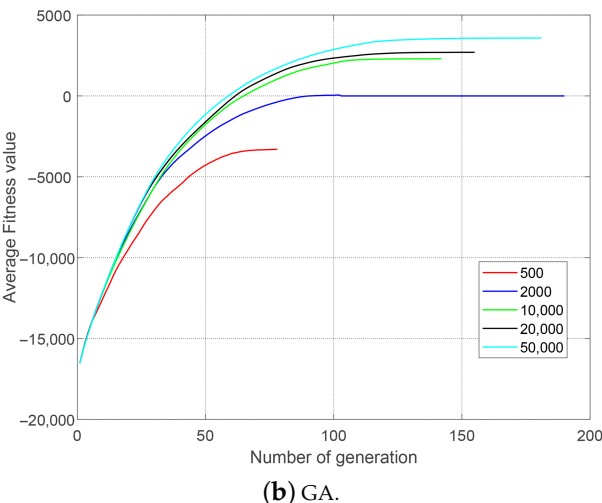

(**b**) GA.

**Figure 4.** Trends of average fitness values of GA-AS and GA with different population sizes.

(**a**) 500

(**b**) 2,000

(**c**) 10,000

(**d**) 50,000

**Figure 5.** Task execution probability of GA-AS and GA with different population sizes.

### 4.3. Analysis of Complexity

As can be seen from Algorithm 1, the GA requires several iterations of computation under the condition that the population size is $NP$, so the time complexity of it is approximated as $O(n^3)$ and the space complexity is approximated as $O(n^2)$. Compared with GA, GA-AS differs mainly in its initial population generation method, which is shown in Algorithm 2, so its time and space complexity can be approximated as $O(n^3)$ and $O(n^2)$, respectively. Since the complexity of the improved A* strategy is smaller than that of the GA, the time and space complexity of GA-AS remains the same as the GA.

The running times of GA and GA-AS with the same computer configuration are shown in Table 3. As can be seen from Table 3, the running time of GA-AS is almost always larger than that of GA for the same population size, and this extra time can be considered as the running time of the improved A* strategy.

## 5. Discussion

In this work, the LEO satellite network using ISLL with stochastic link failure is modeled, in which the constraints and objective functions are given. Then, we propose an improved GA called GA-AS to optimize the model with the indicators of task revenue, switching times and routing cost in the case of the stochastic link failure. For reducing the cost of computation, a new initial solution generator is presented by combining the A* with the roulette wheel selection strategy. Finally, experimental results show that the GA-AS can significantly reduce the cost of computation and improve the optimization efficiency compared to GA.

**Author Contributions:** Data curation, G.Z.; Formal analysis, G.Z.; Funding acquisition, S.W.; Investigation, G.Z.; Methodology, G.Z. and Z.K.; Software, G.Z.; Supervision, Z.K. and S.W.; Writing—original draft, G.Z.; Writing—review and editing, G.Z., Z.K. and Y.H. All authors have read and agreed to the published version of the manuscript.

**Funding:** This research was funded by the National Natural Science Foundation of China, grant number [U20B2056].

**Institutional Review Board Statement:** Not applicable.

**Informed Consent Statement:** Informed consent was obtained from all subjects involved in the study.

**Data Availability Statement:** The data presented in this study can be made available on request from the corresponding author.

**Acknowledgments:** The authors are grateful to the Shanghai Jiao Tong University, in particular to The school of Aeronautics and Astronautics, for their support and assistance.

**Conflicts of Interest:** The authors declare no conflicts of interest.

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
