# Peer review of "A Routing Optimization Method for LEO Satellite Networks with Stochastic Link Failure"

_aerospace, doi:10.3390/aerospace9060322_

Round 1

Reviewer 1 Report

This papers deals with an interesting and timely topic, since LEO mega-constellations will likely be part of future B5G and 6G networks. The proposed solution is interesting and the numerical results show a good performance.

However, there are several aspects that the authors shall address, which are reported below.

Detailed comments

The major drawback that I find in this work is related to the assumptions on the operational constraints at page 5. In particular:

  • what does "disregard the power limit of the satellite" mean? On-board power consumption and limits are one of the major challenges/issues in SatCom, so this assumption needs to be deeply justified and clarified;
  • during the considered period, the number of satellites is assumed to be constant. This is a strong assumption, since we are talking about LEO constellations. If global coverage is needed, this assumption will not hold because the satellites in visibility will for sure never be the same, or in the same number. If no global coverage is needed, this can be guaranteed with proper orbital manoeuvring and control of the satellites, which makes the constellation a bit more complex to be managed. These aspects need a thorough clarification.

An analysis of the computational complexity and the convergence of the proposed algorithm shall be provided. No information is reported in the paper related to these important aspects.

On page 2, the authors state that "The space segment mainly contains satellites near the earth. The ground segment mainly includes ground networks such as gateway stations, and the user segment is the objects served by the satellite network." Why "mainly"? If we are talking about the system architecture for this paper, then the authors can report what are the elements in each segment, without any ambiguity or leaving something outside. If the authors are talking in general about these segments, then all of the elements shall be listed and then those considered in this paper shall be clearly reported. The system architecture needs to be clear and unambiguous.

Some figures, as Fig. 1 for instance, are very small...it would be better to enlarge them a bit to make them readable without the need to significantly zoom in.

Finally, the quality of the written text needs to be significantly improved, due to unclear sentences or wrong English grammar. Moreover, there are several typos to be corrected, e.g., on page 1 MEO is Medium Earth Orbit, not Middle-Earth. 

Author Response

 Dear Associate Reviewer, 

Based on your review, we have carefully revised and checked the article. Please see the attachment for the revised content and responses.

 Guohong Zhao, Zeyu Kang, Yixin Huang, and Shufan Wu 

Reviewer 2 Report

This paper investigates a routing algorithm for LEO satellite networks. Overall, the topic of this paper is interesting. Also, this paper is well-organized. This reviewer suggests the following comments for the acceptance of this paper.

  1. To show the efficacy of the proposed method, the authors need to compare the performance of the proposed method over additional conventional routing algorithms. Only comparison between GA and GA-AS is not enough.

  1. In the simulation part, what is the fitness value? With the current version, it is hard to match the fitness value with the contents of the body part since it is not explicitly presented. It would be better to clarify the definition of the fitness value and describe the details of the result. Why the GA has negative fitness value? Why the range of the fitness value is distinct between GA and GA-AS?

  1. In Figure 5, it seems that the legend of this figure needs to be corrected. It is shown that there are ‘GA’ and ‘GA-PS’. What is the ‘GA-PS’?

  1. As a minor comment, equation (11) needs to be aligned well. Also, the authors need to correct some grammatical errors.

Author Response

(The authors gave the same response as above.)

Round 2

Reviewer 1 Report

The authors addressed the points raised in the previous review iteration and the paper might be accepted as is. Just a few minor editing is suggested for typos.

Reviewer 2 Report

This reviewer has no further comments.